

# Comparison of admission random glucose, fasting glucose, and glycated hemoglobin in predicting the neurological outcome of acute ischemic stroke: a retrospective study

Jia-Ying Sung[1,2], Chin-I Chen[1,2], Yi-Chen Hsieh[3], Yih-Ru Chen[4], Hsin-Chiao Wu[4], Lung Chan[2,5], Chaur-Jong Hu[2,5], Han-Hwa Hu[2,5], Hung-Yi Chiou[4] and Nai-Fang Chi[2,5]

[1] Department of Neurology, Wanfang Hospital, Taipei Medical University, Taipei, Taiwan
[2] Department of Neurology, School of Medicine, College of Medicine, Taipei Medical University, Taipei, Taiwan
[3] The PhD Program of Neural Regenerative Medicine, College of Medical Science and Technology, Taipei Medical University, Taipei, Taiwan
[4] School of Public Health, College of Public Health, Taipei Medical University, Taipei, Taiwan
[5] Department of Neurology, Shuang Ho Hospital, Taipei Medical University, New Taipei City, Taiwan

Corresponding author
Nai-Fang Chi,
naifangchi@tmu.edu.tw

## ABSTRACT

**Background.** Hyperglycemia is a known predictor of negative outcomes in stroke. Several glycemic measures, including admission random glucose, fasting glucose, and glycated hemoglobin (HbA1c), have been associated with bad neurological outcomes in acute ischemic stroke, particularly in nondiabetic patients. However, the predictive power of these glycemic measures is yet to be investigated.

**Methods.** This retrospective study enrolled 484 patients with acute ischemic stroke from January 2009 to March 2013, and complete records of initial stroke severity, neurological outcomes at three months, and glycemic measures were evaluated. We examined the predictive power of admission random glucose, fasting glucose, and HbA1c for neurological outcomes in acute ischemic stroke. Furthermore, subgroup analyses of nondiabetic patients and patients with diabetes were performed separately.

**Results.** Receiver operating characteristic (ROC) analysis revealed that admission random glucose and fasting glucose were significant predictors of poor neurological outcomes, whereas HbA1c was not (areas under the ROC curve (AUCs): admission random glucose = 0.564, $p = 0.026$; fasting glucose = 0.598, $p = 0.001$; HbA1c = 0.510, $p = 0.742$). Subgroup analyses of nondiabetic patients and those with diabetes revealed that only fasting glucose predicts neurological outcomes in patients with diabetes, and the AUCs of these three glycemic measures did not differ between the two groups. A multivariate logistic regression analysis of the study patients indicated that only age, initial stroke severity, and fasting glucose were independent predictors of poor neurological outcomes, whereas admission random glucose and HbA1c were not (adjusted odds ratio: admission random glucose = 1.002, $p = 0.228$; fasting glucose = 1.005, $p = 0.039$; HbA1c = 1.160, $p = 0.076$). Furthermore, subgroup multivariate logistic regression analyses of nondiabetic patients and those with diabetes indicated

that none of the three glycemic measures were associated with poor neurological outcomes.

**Discussion**. Fasting glucose is an independent predictor of poor neurological outcomes in patients with acute ischemic stroke and had greater predictive power than that of admission random glucose and HbA1c. The predictive power of glycemic measures for poor neurological outcomes did not differ significantly between the nondiabetic patients and those with diabetes.

## INTRODUCTION

Hyperglycemia may be present in more than half of patients with acute ischemic stroke (*Scott et al., 1999*). It is associated with an increased risk of infarction volume growth and hemorrhagic transformation (*Paciaroni et al., 2009*; *Parsons et al., 2002*), a decreased recanalization rate of thrombolytic therapy (*Ribo et al., 2005*), and an increased risk of poor neurological outcomes and stroke mortality (*Capes et al., 2001*). Several indices of hyperglycemia have been reported in stroke: elevated fasting glucose (*Yao et al., 2016*), elevated admission random glucose (*Ahmed et al., 2010*; *Ribo et al., 2005*), maximum glucose during the acute disease stage (*Fuentes et al., 2009*), and elevated glycated hemoglobin (HbA1c) (*Kamouchi et al., 2011*; *Lattanzi et al., 2016*). The levels of fasting glucose, admission random glucose, and maximum glucose have been associated with the stress response of stroke, particularly in nondiabetic patients (*Capes et al., 2001*; *Stead et al., 2009*), and HbA1c was considered as a marker for prestroke glycemic control (*Kamouchi et al., 2011*). Moreover, some studies have reported that in acute ischemic stroke, hyperglycemia is more detrimental in nondiabetic patients than in those with diabetes (*Capes et al., 2001*; *Stead et al., 2009*; *Yao et al., 2016*). Although these glycemic measures have been associated with neurological outcomes in acute ischemic stroke, the most predictive factor among these is yet to be reported. The present study compared the predictive performance of admission random glucose, fasting glucose, and HbA1c for poor neurological outcomes in acute ischemic stroke. Furthermore, the predictive performance of these glycemic measures was evaluated through subgroup analysis of patients with diabetes and nondiabetic patients.

## MATERIALS & METHODS

### Study patients

The present study was approved by the Joint Institutional Review Board of Taipei Medical University (No. 201304008). Data were obtained by reviewing medical records of the patients with acute ischemic stroke admitted to the hospitals affiliated with Taipei Medical University, Wan Fang Hospital and Shuang Ho Hospital, between January 2009 and March 2013. Data were included for analysis if the patients were aged ≥20 years, presented to the emergency department with acute ischemic stroke and were subsequently admitted to the neurology department, and complete records of admission random glucose, fasting

glucose, HbA1c, National Institute of Health Stroke Scale (NIHSS) scores at admission, and modified Rankin Scale (mRS) scores at three months after stroke were obtained. After reviewing the enrolled medical records, patients with a history of stroke or who died or were lost to follow-up within three months after stroke were excluded.

## Definitions of clinical characteristics and neurological outcomes

In the present study, acute ischemic stroke was confirmed by clinical symptoms and neuroimages. The etiologic subtype of acute ischemic stroke was classified according to the Trial of Org 10172 in Acute Stroke Treatment (TOAST) study (*Adams et al., 1993*). Admission random glucose was the first nonfasting plasma glucose measured at the emergency department. Fasting plasma glucose and HbA1c were measured on the first morning of admission after at least 8 h of fasting. Hypertension was defined as a systolic blood pressure >140 mmHg, a diastolic blood pressure >90 mmHg, or current use of antihypertensive medicines. Diabetes was defined as HbA1c $\geq$ 6.5% or current use of antidiabetic medicines. Hyperlipidemia was defined as total cholesterol >200 mg/dL or current use of cholesterol-lowering medicines. Atrial fibrillation was diagnosed according to either a documented history or an electrocardiogram on admission. Initial stroke severity was evaluated on the first day of admission on the basis of the NIHSS at admission, and neurological outcomes were evaluated on the basis of the mRS at three months after stroke. Poor neurological outcomes were defined as an mRS score = 3, 4, or 5 at three months after stroke.

## Statistical analyses

The continuous variables, rank variables, and proportions of categorical variables were compared between the nondiabetic patients and those with diabetes by using the Student $t$, Mann–Whitney $U$, and chi-squared tests, respectively. Receiver operating characteristic (ROC) analysis was conducted for evaluating the predictive performance of the glycemic measures, namely admission random glucose, fasting glucose, and HbA1c, in distinguishing between the favorable and poor neurological outcomes. Specifically, the predictive performance of these glycemic measures was evaluated by comparing their areas under the ROC curve (AUCs) (*DeLong, DeLong & Clarke-Pearson, 1988*). A univariate logistic regression analysis was conducted to estimate the odds ratio (OR) of poor neurological outcomes for each variable. Age, sex, and variables with $p \leq 0.2$ in the univariate logistic regression analysis were included in a multivariate logistic regression model to adjust the OR of poor neurological outcomes for the glycemic measures. Parametric and nonparametric data were expressed as the mean $\pm$ standard deviation (SD) and as the median with the interquartile range (IQR), respectively, and $p < 0.05$ was considered statistically significant. Statistical data were analyzed using PASW Statistics for Windows version 18.0 (SPSS Inc., Chicago, IL, USA) and MedCalc Statistical Software version 16.8 (MedCalc Software bvba, Ostend, Belgium; https://www.medcalc.org, 2016).

## RESULTS

Table 1 summarizes the clinical characteristics of the study patients. Of the 484 patients enrolled in the present study, 212 were nondiabetic and 272 had diabetes. Age, the

**Table 1** Clinical characteristics of the patients.

| | Total patients (n = 484) | Nondiabetic patients (n = 212) | Diabetic patients (n = 272) | p value |
|---|---|---|---|---|
| Age, mean ± SD, years | 67.9 ± 13.7 | 65.9 ± 15.3 | 69.5 ± 12.2 | 0.004* |
| Male sex, n (%) | 315 (65.1%) | 145 (68.4%) | 170 (62.5%) | 0.177 |
| Comorbidities, n (%) | | | | |
| Hypertension | 346 (71.5%) | 132 (62.3%) | 214 (78.7%) | <0.001* |
| Diabetes | 272 (56.2%) | 0 (0%) | 212 (100%) | N.A. |
| Hyperlipidemia | 147 (30.4%) | 62 (29.2%) | 85 (31.2%) | 0.635 |
| Atrial fibrillation | 74 (15.3%) | 38 (17.9%) | 36 (13.2%) | 0.155 |
| NIHSS at admission, median (IQR) | 4 (2–8) | 4 (2–8) | 4 (2–7) | 0.972 |
| mRS at 3 months, median (IQR) | 1 (1–2) | 1 (1–2) | 1 (1–3) | 0.654 |
| Poor neurological outcome, n (%) | 118 (24.4%) | 48 (22.6%) | 70 (25.7%) | 0.432 |
| Stroke etiology subtypes, n (%) | | | | 0.658 |
| Large artery atherosclerosis | 133 (27.5%) | 56 (26.4%) | 77 (28.3%) | |
| Small artery occlusion | 166 (34.3%) | 69 (32.5%) | 97 (35.7%) | |
| Cardioembolism | 26 (5.4%) | 12 (5.7%) | 14 (5.1%) | |
| Other determined etiology | 1 (0.2%) | 1 (0.5%) | 0 (0%) | |
| Undremined etiology | 158 (32.6%) | 74 (34.9%) | 84 (30.9%) | |
| Admission random glucose, mean ± SD, mg/dL | 175.5 ± 86.8 | 129.5 ± 34.8 | 211.4 ± 97.6 | <0.001* |
| Fasting glucose, mean ± SD, mg/dL | 132.9 ± 58.7 | 102.1 ± 19.9 | 156.9 ± 67.3 | <0.001* |
| HbA1c, mean ± SD, % | 7.0 ± 1.8 | 5.8 ± 0.4 | 8.0 ± 1.8 | < 0.001* |

Notes.
*$p < 0.05$.
IQR, interquartile range; SD, standard deviation.

proportion of hypertension, admission random glucose, and fasting glucose in the patients with diabetes were significantly higher than those in the nondiabetic patients, and the NIHSS score at admission, mRS score at three months after stroke, and proportions of poor neurological outcomes and the stroke etiology subtype did not differ between the two groups.

The AUCs of admission random glucose, fasting glucose, and HbA1c in all patients are presented in Fig. 1A. In all patients, admission random glucose and fasting glucose were significant predictors of poor neurological outcomes (AUCs: admission random glucose = 0.564, 95% confidence interval (CI) [0.519–0.609], $p = 0.026$; fasting glucose = 0.598, 95% CI [0.553–0.642], $p = 0.001$); however, HbA1c was not a significant predictor (AUC = 0.510, 95% CI [0.465–0.556], $p = 0.742$). The cutoff values with the highest sensitivity and specificity were 127 mg/dL for admission random glucose (sensitivity = 74.6%, specificity = 39.3%) and 109 mg/dL for fasting glucose (sensitivity = 67.8%, specificity = 51.6%). The AUCs of admission random glucose and fasting glucose did not differ significantly (0.564 vs. 0.598, $p = 0.194$), and the AUCs of admission random glucose and fasting glucose were significantly higher than that of HbA1c (0.564 vs. 0.510, $p = 0.038$ and 0.598 vs. 0.510, $p = 0.001$, respectively). The ROC curves of admission random glucose, fasting glucose, and HbA1c in the nondiabetic patients ($n = 212$) and patients with diabetes ($n = 272$) are presented in Figs. 1B and 1C, respectively. In the nondiabetic patients, fasting glucose was not a significant predictor of poor neurological outcomes (AUC = 0.593, 95% CI

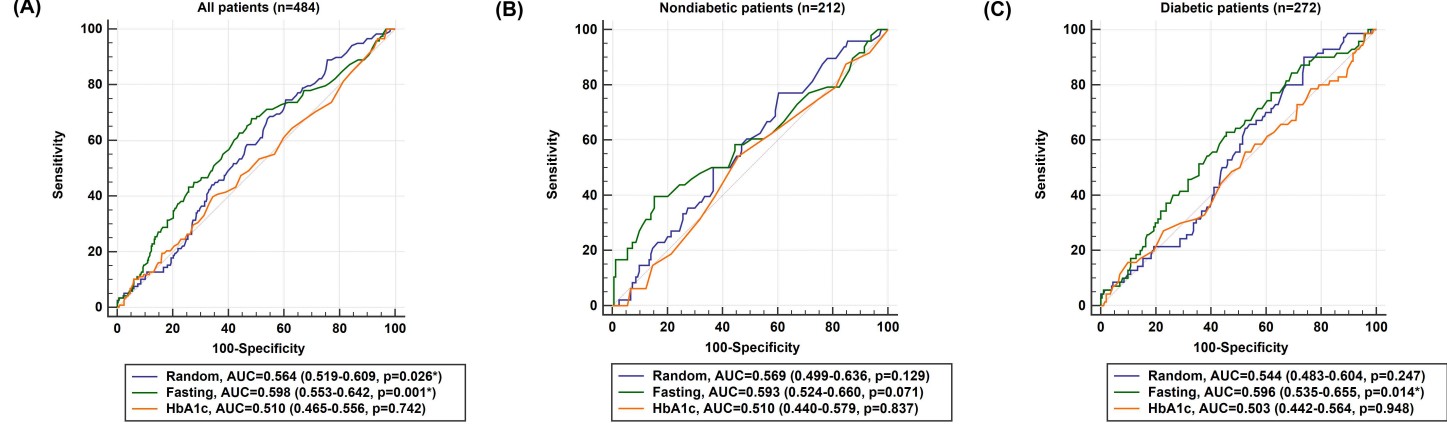

**Figure 1** Receiver operating characteristic (ROC) curves of admission random glucose, fasting glucose, and HbA1c for predicting poor neurological outcomes (modified Rankin Scale = 3, 4, or 5 at 3 months) in (A) all study patients, (B) nondiabetic patients, and (C) patients with diabetes. $^*p < 0.05$.

[0.524–0.660], $p = 0.071$) with an optimal cutoff value of 109 mg/dL (sensitivity = 39.6%, specificity = 84.8%), whereas admission random glucose and HbA1c were nonsignificant predictors of poor neurological outcomes (AUCs: admission random glucose = 0.569, 95% CI [0.499–0.636], $p = 0.129$; HbA1c = 0.510, 95% CI [0.440–0.579], $p = 0.837$). In the patients with diabetes, fasting glucose was a significant predictor of poor neurological outcomes (AUC = 0.596, 95% CI [0.535–0.655], $p = 0.014$) with an optimal cutoff value of 140 mg/dL (sensitivity = 62.9%, specificity = 54.5%), whereas admission random glucose and HbA1c were nonsignificant predictors of poor neurological outcomes (AUCs: admission random glucose = 0.544, 95% CI [0.483–0.604], $p = 0.247$; HbA1c = 0.503, 95% CI [0.442–0.564], $p = 0.948$). The AUCs of glycemic measures were not significantly different between the nondiabetic patients and those with diabetes (AUCs: admission random glucose = 0.569 vs. 0.544, $p = 0.671$; fasting glucose = 0.593 vs. 0.596, $p = 0.963$; HbA1c = 0.510 vs. 0.503, $p = 0.911$).

Table 2 presents the results of univariate and multivariate logistic regression analyses of variables potentially predicting poor neurological outcomes. In univariate logistic regression analysis, age, the NIHSS score at admission, atrial fibrillation, and fasting glucose were significant predictors of poor neurological outcomes in the study patients (crude OR for fasting glucose = 1.005, 95% CI [1.002–1.008], $p = 0.003$). However, neither admission random glucose nor HbA1c were significant predictors of poor neurological outcomes. The adjusted OR of admission random glucose was determined using multivariate logistic regression model 1, which included age, sex, the NIHSS score at admission, atrial fibrillation, and admission random glucose. Age and the NIHSS score at admission were significant predictors of poor neurological outcomes, and admission random glucose was not associated with poor neurological outcomes in model 1 (adjusted OR = 1.002, 95% CI [0.999–1.005], $p = 0.228$). The adjusted OR for fasting glucose was determined using multivariate logistic regression model 2, which included age, sex, the NIHSS score at admission, atrial fibrillation, and fasting glucose. Age, the NIHSS score at admission, and fasting glucose

**Table 2  Univariate and multivariate logistic regression analyses of clinical characteristics and glycemic measures to poor neurological outcome.**

| Characteristics | Crude OR (95% CI) | p value | Adjusted OR model 1 (95% CI) | p value | Adjusted OR model 2 (95% CI) | p value | Adjusted OR model 3 (95% CI) | p value |
|---|---|---|---|---|---|---|---|---|
| | | | Poor neurological outcome (mRS $\geq$ 3 at three months) | | | | | |
| | | | All patients ($n = 484$) | | | | | |
| Age | 1.059 (1.040–1.080) | <0.001[*] | 1.056 (1.032–1.080) | <0.001[*] | 1.059 (1.034–1.084) | <0.001[*] | 1.059 (1.034–1.084) | <0.001[*] |
| Male sex | 0.755 (0.492–1.159) | 0.199 | 0.816 (0.472–1.411) | 0.467 | 0.811 (0.468–1.404) | 0.453 | 0.833 (0.481–1.443) | 0.515 |
| NIHSS at admission | 1.320 (1.246–1.398) | <0.001[*] | 1.313 (1.237–1.395) | <0.001[*] | 1.312 (1.235–1.392) | <0.001[*] | 1.319 (1.241–1.402) | <0.001[*] |
| Hypertension | 1.159 (0.726–1.851) | 0.535 | | | | | | |
| Hyperlipidemia | 0.768 (0.482–1.223) | 0.266 | | | | | | |
| Atrial fibrillation | 2.870 (1.710–4.817) | 0.001[*] | 1.387 (0.679–2.833) | 0.370 | 1.373 (0.668–2.822) | 0.388 | 1.452 (0.706–2.986) | 0.311 |
| Admission random glucose | 1.002 (0.999–1.004) | 0.158 | 1.002 (0.999–1.005) | 0.228 | | | | |
| Fasting glucose | 1.005 (1.002–1.008) | 0.003[*] | | | 1.005 (1.003–1.010) | 0.039[*] | | |
| HbA1c | 1.032 (0.919–1.158) | 0.600 | | | | | 1.160 (0.985–1.366) | 0.076 |
| | | | Nondiabetic patients ($n = 212$) | | | | | |
| Age | 1.0061 (1.033–1.089) | <0.001[*] | 1.065 (1.027–1.103) | 0.006[*] | 1.067 (1.028–1.106) | 0.006[*] | 1.066 (1.028–1.106) | 0.006[*] |
| Male sex | 0.632 (0.323–1.233) | 0.178 | 0.722 (0.292–1.782) | 0.479 | 0.711 (0.288–1.758) | 0.461 | 0.685 (0.275–1.702) | 0.415 |
| NIHSS at admission | 1.347 (1.231–1.473) | <0.001[*] | 1.355 (1.229–1.493) | <0.001[*] | 1.345 (1.220–1.482) | <0.001[*] | 1.359 (1.232–1.500) | <0.001[*] |
| Hypertension | 1.013 (0.521–1.968) | 0.969 | | | | | | |
| Hyperlipidemia | 0.659 (0.311–1.394) | 0.275 | | | | | | |
| Atrial fibrillation | 2.787 (1.313–5.916) | 0.008[*] | 1.424 (0.498–4.076) | 0.510 | 1.334 (0.471–3.792) | 0.586 | 1.396 (0.494–5.859) | 0.529 |
| Admission random glucose | 1.003 (0.995–1.012) | 0.453 | 0.999 (0.987–1.012) | 0.886 | | | | |
| Fasting glucose | 1.025 (1.009–1.041) | 0.002[*] | | | 1.011 (0.992–1.031) | 0.272 | | |
| HbA1c | 1.124 (0.493–2.564) | 0.782 | | | | | 1.700 (0.494–5.859) | 0.400 |

**Table 2** (*continued*)

| Characteristics | Crude OR (95% CI) | p value | Adjusted OR model 1 (95% CI) | p value | Adjusted OR model 2 (95% CI) | p value | Adjusted OR model 3 (95% CI) | p value |
|---|---|---|---|---|---|---|---|---|
| | | | **Poor neurological outcome (mRS ≥ 3 at three months)** | | | | | |
| | | | **Diabetic patients (n = 272)** | | | | | |
| Age | 1.057 (1.030–1.086) | <0.001* | 1.049 (1.017–1.082) | 0.002* | 1.053 (1.021–1.087) | 0.001* | 1.055 (1.021–1.089) | 0.001* |
| Male sex | 0.867 (0.497–1.514) | 0.616 | 0.892 (0.445–1.788) | 0.747 | 0.873 (0.434–1.755) | 0.703 | 0.914 (0.455–1.837) | 0.801 |
| NIHSS at admission | 1.304 (1.209–1.407) | <0.001* | 1.291 (1.194–1.395) | <0.001* | 1.286 (1.190–1.389) | <0.001* | 1.295 (1.197–1.400) | <0.001* |
| Hypertension | 1.257 (0.632–2.500) | 0.515 | | | | | | |
| Hyperlipidemia | 0.843 (0.464–1.531) | 0.575 | | | | | | |
| Atrial fibrillation | 3.089 (1.500–6.361) | 0.002* | 1.436 (0.527–3.915) | 0.480 | 1.361 (0.493–3.761) | 0.552 | 1.496 (0.545–4.111) | 0.435 |
| Admission random glucose | 1.001 (0.999–1.004) | 0.319 | 1.002 (0.998–1.006) | 0.319 | | | | |
| Fasting glucose | 1.004 (1.001–1.008) | 0.003* | | | 1.005 (0.999–1.010) | 0.110 | | |
| HbA1c | 1.000 (0.863–1.161) | 0.992 | | | | | 1.163 (0.945–1.430) | 0.154 |

**Notes.**

*$p < 0.05$.

OR, odds ratio.

Model 1 included age, sex, NIHSS at admission, atrial fibrillation, and admission random glucose.

Model 2 included age, sex, NIHSS at admission, atrial fibrillation, and fasting glucose.

Model 3 included age, sex, NIHSS at admission, atrial fibrillation, and HbA1c.

were significant predictors of poor neurological outcomes in model 2 (adjusted OR for fasting glucose = 1.005, 95% CI [1.003–1.010], $p = 0.039$). The adjusted OR of HbA1c was calculated using multivariate logistic regression model 3, which included age, sex, the NIHSS score at admission, atrial fibrillation, and HbA1c. Age and the NIHSS score at admission were significant predictors of poor neurological outcomes, and HbA1c was not associated with poor neurological outcomes in model 3 (adjusted OR = 1.160, 95% CI [0.985–1.366], $p = 0.076$).

Subgroup univariate and multivariate logistic regression analyses were performed in the nondiabetic patients and patients with diabetes (Table 2). In the univariate logistic regression analysis, age, the NIHSS score at admission, atrial fibrillation, and fasting glucose were significant predictors of poor neurological outcomes (crude OR for fasting glucose: nondiabetic patients = 1.025, 95% CI [1.009–1.041], $p = 0.002$; patients with diabetes = 1.004, 95% CI [1.001–1.008], $p = 0.003$). However, after adjustment for the influence of age, sex, the NIHSS score at admission, and atrial fibrillation, all three glycemic measures were not associated with poor neurological outcomes in both groups.

## DISCUSSION

In the present study, of the three glycemic measures, fasting glucose was an independent predictor and the most predictive factor of poor neurological outcomes in patients with acute ischemic stroke. ROC analysis revealed that admission random glucose was a predictor of poor neurological outcomes in all study patients; however, it was not associated with poor neurological outcomes in the logistic regression analysis. Furthermore, ROC and logistic regression analyses revealed that HbA1c was not a predictive factor of poor neurological outcomes. The predictive value of glycemic measures for poor neurological outcomes did not differ significantly between the nondiabetic patients and patients with diabetes.

ROC analysis revealed that fasting glucose and admission random glucose have more predictive value than HbA1c does for poor neurological outcomes in acute ischemic stroke. Studies have reported that a high level of fasting or random glucose is associated with poor neurological outcomes or mortality in acute ischemic stroke (*Capes et al., 2001*; *Fang et al., 2013*; *Hu et al., 2012*), and hyperglycemia at the acute stage of the disease may be largely caused by stress (*Douketis, 2002*). By contrast, HbA1c represents average glycemic levels in the preceding 6–8 weeks and is not affected by transient hyperglycemia; therefore, HbA1c has been perceived as a marker for prestroke glycemic control (*Kamouchi et al., 2011*; *Lattanzi et al., 2016*). Furthermore, HbA1c has been associated with increased mortality (*Wu et al., 2014*) and poor neurological outcomes (*Gao et al., 2016*; *Hjalmarsson et al., 2014*; *Lattanzi et al., 2016*) in acute ischemic stroke; however, it has also been reported to be nonpredictive of neurological outcomes (*Shin et al., 2015*). Although many studies have reported the role of hyperglycemia in predicting poor neurological outcomes in acute ischemic stroke, the inclusion criteria for patients, definition of hyperglycemia, and outcome measures varied between these studies; therefore, the results are difficult to compare. Moreover, no study has compared the predictive performance of different definitions of hyperglycemia for neurological outcomes. The results of our study revealed that stress hyperglycemia contributed more than prestroke glycemic control did in predicting negative outcomes in acute ischemic stroke. The ROC analysis revealed that the AUCs of fasting glucose and admission random glucose did not differ significantly; nevertheless, the AUC of fasting glucose was the highest among the three glycemic measures. Furthermore, multivariate logistic regression analysis revealed that fasting glucose was the only independent predictor of poor neurological outcomes among the three glycemic measures (Table 2). Therefore, we speculate that fasting glucose is superior to admission random glucose in predicting neurological outcomes. However, why fasting glucose was more predictive than admission random glucose in this study remains uncertain, and further investigations are warranted.

Some studies have reported that in acute ischemic stroke, hyperglycemia is more detrimental in nondiabetic patients than in patients with diabetes (*Capes et al., 2001*; *Stead et al., 2009*; *Yao et al., 2016*); however, the present findings are not consistent with the aforementioned results, possibly because of the differences in study designs, statistical methods, definition of hyperglycemia, and functional outcome measures. Most studies have used a cutoff value of blood glucose as the criterion for defining hyperglycemia, such as fasting

glucose >140 mg/dL (*Woo et al., 1990*) or nonfasting glucose >130 mg/dL (*Stead et al., 2009*). However, we used ROC analysis instead of a cutoff value because of the lack of a widely accepted criterion for defining hyperglycemia in acute ischemic stroke. Moreover, ROC analysis enables the comparison of the predictive performance of different glycemic measures and provides information on sensitivity and specificity, which cannot be obtained by defining hyperglycemia on the basis of a cutoff value.

In preclinical animal studies, controlling hyperglycemia with insulin resulted in heterogeneous neuroprotection outcomes (*MacDougall & Muir, 2011*). In randomized controlled trials of intensive insulin treatment for glycemic control in patients with acute ischemic stroke, there was no difference in death or dependency between the treatment and control groups, and the results were the same in subgroup analyses of patients with diabetes and nondiabetic patients (*Bellolio, Gilmore & Ganti, 2014*). Therefore, the pathophysiology of hyperglycemia in acute ischemic stroke probably involves other factors in addition to high blood glucose. Stress-related cytotoxic neurotransmitters, inflammation, or insulin resistance could be associated with an increased risk of poor neurological outcomes and mortality in patients with stroke and hyperglycemia (*Arenillas, Moro & Davalos, 2007*; *Harada, Fujita-Hamabe & Tokuyama, 2012*; *Kruyt et al., 2010*). The clinical guidelines published by the American Heart Association/American Stroke Association and the European Stroke Organization recommend that the glycemic levels of patients with acute ischemic stroke be maintained below 180 mg/dL. This recommendation is based on the consensus of experts because evidence does not support maintaining the blood glucose at a specific level improves outcomes (*European Stroke Organisation Executive & Committee, 2008*; *Jauch et al., 2013*). Therefore, although hyperglycemia has been confirmed to be a predictor of negative outcomes in stroke, additional investigations are warranted for elucidating the underlying mechanisms and treatment strategies.

The present study had some limitations. First, most patients enrolled had mild to moderate initial stroke severity (median NIHSS score at admission = 4, IQR = 2–8), and the influence of hyperglycemia might be more prominent in patients with severe stroke. However, only 84 patients in the present study had an NIHSS score at admission ≥10, and a subgroup analysis of these patients would not have yielded sufficient statistic power. Second, mortality was not included as an outcome because the timing of and reason for mortality could not be assessed accurately in the retrospective medical review. Some patients might have died or received follow-up in facilities other than the two hospitals included in the present study; therefore, the patients who died or were lost to follow-up within three months after stroke were not included in this study. Because of the aforementioned limitations, the results of this study cannot be generalized to all ischemic stroke patients. Third, diabetes was defined as HbA1c < 6.5% or as medication usage; therefore, patients with diabetes who were on dietary modification alone with satisfactory HbA1c may have been misclassified as nondiabetic patients.

## CONCLUSIONS

Fasting glucose is an independent predictor of poor neurological outcomes in patients with acute ischemic stroke, with more predictive power than that of admission random glucose

and HbA1c. The predictive value of glycemic measures for poor neurological outcomes in acute ischemic stroke did not differ significantly between the nondiabetic patients and those with diabetes.

### Funding

This work was supported by a grant from Taipei Medical University (98 TMU-SHH-04-03), Ministry of Science and Technology, Taiwan (104-2314-B-038-012-MY3, 104-2314-B-038-023-), Ministry of Health and Welfare (MOHW105-TDU-B-212-133018), and in part by The Taiwan Ministry of Health and Welfare Clinical Trial and Research Center of Excellence (MOHW105-TDU-B-212-133019), the China Medical University Hospital, the Academia Sinica Taiwan Biobank, the Stroke Biosignature Project (BM10501010037), and the NRPB Stroke Clinical Trial Consortium (MOST 104-2325- B-039 - 005-). All the funding or sources of support were received during the period of this study. There was no additional external funding received for this study. The funders had no role in study design, data collection and analysis, decision to publish, or preparation of the manuscript.

### Grant Disclosures

The following grant information was disclosed by the authors:
Taipei Medical University: 98 TMU-SHH-04-03.
Ministry of Science and Technology, Taiwan: 104-2314-B-038-012-MY3, 104-2314-B-038-023-.
Ministry of Health and Welfare: MOHW105-TDU-B-212-133018.
Taiwan Ministry of Health and Welfare Clinical Trial and Research Center of Excellence: MOHW105-TDU-B-212-133019.
China Medical University Hospital, Academia Sinica Taiwan Biobank: BM10501010037.
NRPB Stroke Clinical Trial Consortium: MOST 104-2325- B-039 - 005-.

### Competing Interests

The authors declare there are no competing interests.

### Author Contributions

- Jia-Ying Sung conceived and designed the experiments, performed the experiments, analyzed the data, wrote the paper, reviewed drafts of the paper.
- Chin-I Chen conceived and designed the experiments, performed the experiments, wrote the paper, reviewed drafts of the paper.
- Yi-Chen Hsieh conceived and designed the experiments, analyzed the data, reviewed drafts of the paper.
- Yih-Ru Chen conceived and designed the experiments, performed the experiments, analyzed the data.
- Hsin-Chiao Wu performed the experiments, analyzed the data, contributed reagents/materials/analysis tools, prepared figures and/or tables.
- Lung Chan wrote the paper.

- Chaur-Jong Hu wrote the paper, reviewed drafts of the paper.
- Han-Hwa Hu conceived and designed the experiments, reviewed drafts of the paper.
- Hung-Yi Chiou conceived and designed the experiments, wrote the paper, reviewed drafts of the paper.
- Nai-Fang Chi conceived and designed the experiments, performed the experiments, analyzed the data, contributed reagents/materials/analysis tools, wrote the paper, prepared figures and/or tables, reviewed drafts of the paper.

## Human Ethics

The following information was supplied relating to ethical approvals (i.e., approving body and any reference numbers):

Joint Institutional Review Board of Taipei Medical University (No. 201304008).

## Data Availability

The raw data has been supplied as Data S1.

## Supplemental Information

Supplemental information for this article can be found online at http://dx.doi.org/10.7717/peerj.2948#supplemental-information.

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
