# Peer review of "Comparison of admission random glucose, fasting glucose, and glycated hemoglobin in predicting the neurological outcome of acute ischemic stroke: a retrospective study"

_PeerJ, doi:10.7717/peerj.2948_

## Round 0.1 · original submission · Major Revisions

Dear Authors,

Please do the necessary revisions that are the major hurdles to the manuscript being accepted for publication.

Reviewer 1 ·

Basic reporting

No Comments

Experimental design

1. It is a retrospective study, but still has large & adequate number of subjects for analysis.
2. Due to the type of study, there was a limitation in terms of investigations performed. However the authors had selected subjects that had complete set of tests.

3.The diagnosis of diabetics was based on HbA1C <6.5% or on medications, thus it may excluded diabetic patients who were on dietary modification alone with good HbA1C.

Validity of the findings

Patients who died or had incomplete test results were excluded from the study, thus the neurological outcome of these patients were not assessed. However the authors had discussed that as part of the limitation of the study.

Additional comments

Eventhough it is a retrospective study and has some limitations, but it contributes to the current knowledge of hyperglycaemia in acute ischemic stroke. The knowledge gap was clearly stated.Thus, it opens for further prospective study by other researchers.

Reviewer 2 ·

Basic reporting

This study was a retrospective study. The study rationale is appropriate. The written language style is clear and easily readable, with appropriate referencing.

Experimental design

The authors looked at predictors of poor neurological outcome between the two groups, using multivariate and univariate analysis and ROC. The experimental design and methods are appropriate for the research question.
Why did the authors exclude history of previous stroke?
What is the rationale for comparing diabetics and non-diabetics? It is expected that diabetics are more likely to have higher fasting, HbAIC and random glucose.

Validity of the findings

The sample size is reasonable.
The predictors of poor neurological outcome using blood glucose measurements alone is not sufficient in clinical practice, as there are many other confounders of poor outcome. Were the other confounders adjusted in these groups?
The ROC curve analysis is not robust, in that the AUC was only in the range of 0.54 - 0.66 for the three parameters tested.
The explanation that fasting glucose is better than random glucose in predicting neurological outcome is not scientifically sound.

Additional comments

The language can be improved with minor editing.

·

Basic reporting

This study sets out to compare the predictive effect of various measures of hyperglycaemia on stroke outcome, with subgroup analysis of diabetic vs. nondiabetic populations.
The authors’ grasp of these concepts is sound and they are well argued. The use of AUC is an improvement over cut-off values to define hyperglycaemia.
The conclusions are potentially useful in clinical practice.
The results are an important addition to the body of knowledge about the behaviour of blood glucose in stroke. Perhaps the authors could expand on their views on the pathogenesis of hyperglycaemia in stroke as well as its role in stroke outcome. Concerning the two negative predictors namely fasting glucose and atrial fibrillation, analysis for possible association might be revealing.

Experimental design

The author’s statement on Limitations of the study is correct: “it is possible that the
218 influence of hyperglycemia would be more prominent in patients with severe stroke.”

This is a crucial decision in the study design. By excluding patients who died, the relative roles of the three glycemic measures may have been spuriously obscured. Those patients who died of stroke are precisely the “severe stroke” group whose larger effect size might allow negative outcome predictors to be revealed by the study.

I recommend that the patients who died should be included in the analysis, while making allowance for limitations in ascertainment of cause of death.

Validity of the findings

EXAMPLES OF STATEMENTS THAT NEED RESTATING OF THE FACTS
1. Line 97:
97: The age,
98 proportion of hypertension, admission random glucose, fasting glucose, and HbA1c in diabetic
99 patients were significant higher than those in nondiabetic patients,
- HbA1c should be omitted; it is higher by definition in the diabetic group.
2. Line 141:
“it showed that age, NIHSS” is re-stated (double mention). Also, this entire sentence could be broken into two parts, like many other long sentences in this paper.

Additional comments

Much of the writing is difficult to follow due to unclear expressions and circular statements. The result is lack of clarity in presentation of findings and discussion of results.
It is necessary to tighten the language used in the entire paper. Nonstandard use of English idioms needs to be corrected (e.g. line 171 “in the contrary” for “on the contrary”). Recasting of the writing by a language expert would be helpful.
EXAMPLES OF LANGUAGE WEAKNESSES
1. Background: “Hyperglycemia is a known detrimental factor of stroke.” – the intended meaning is along the lines of “Hyperglycaemia is a known predictor of negative outcome in stroke”.
2. Line 158: “we found that in the three glycemic measures” – should be “of the three glycemic measures”
3. Line 162:
The HbA1c is not predictive of poor neurological outcome in both ROC curve analyses and logistic regression. – should be “either” and “or”
4. Line 163:
“The predictive performance of glycemic measures
164 in poor neurological outcome is not significantly different”
- should be “… predictive value of glycemic measures for poor neurological outcome…”
5. Line 205:
“involve mechanisms more than high blood glucose only” – should be “mechanisms beyond high blood glucose”.

---

## Round 0.2 · Minor Revisions

Dear Authors, there are still minor revisions needed as suggested by the peer reviewers.Please proceed to do this and get a professional English editor to improve the manuscript grammer and syntax as well

Reviewer 1 ·

Basic reporting

No comment.

Experimental design

1. Line 110 : I suggest to define poor neurological outcomes as an "mRS score ≧ 3 < 6" at 3 months after stroke , as patients with mRS score 6 (death) were excluded from the study.
2. Line 375 and Figure 1 (legend) :Please consider changing to "mRS score ≧ 3 < 6" at 3 months

Validity of the findings

1. I suggest the authors to mention in a clear manner (in the abstract, method, results, discussion and conclusion) that the study was performed in a subgroup of mild ischemic stroke (based on the results of median NIHSS of 4 and median mRS score of 1 ) and only 24.4% of patients had mRS ≥ 3 < 6.
2. The patients who died were excluded in this study, and if they were to be included, their neurological outcome would be mRS score of 6 (death) at 3 months follow up. Thus, the results of this study cannot be generalized to all ischemic stroke patients and it has to be made clear to the readers.

3.Line 150 & 151: p value of 0.071 is not significant.
Please consider changing the sentence to "In the nondiabetic patients, fasting glucose was not a significant predictor of poor neurological outcome.

4. Line 225 , 226,227 : I suggest to delete the whole sentences “We conjecture ..........", as this postulation is not medically sound. Can you try to relate it to the effect of other stress hormones during fasting state or to mention that it requires further research/investigation.

Reviewer 2 ·

Basic reporting

No comment. The article conforms to the standards of the journal

Experimental design

The research is original, with well defined research question. The authors have made the necessary corrections

Validity of the findings

No comments. Appropriate explanation given

Additional comments

The revised manuscript now reads better, with better explanation of the results.

·

Basic reporting

No comment

Experimental design

No comment

Validity of the findings

No comment

Additional comments

The revision is satisfactory. I have only one comment - line 37. Please see comment box in pdf file enclosed.

---

## Round 0.3 · accepted · Accept

The manuscript has been thoroughly reviewed after two revisions and is now accepted for publication in PeerJ

Reviewer 1 ·

Basic reporting

No comment

Experimental design

No comment

Validity of the findings

No comment

Additional comments

The revision is satisfactory.